# Considering *Caenorhabditis elegans* Aging on a Temporal and Tissue Scale: The Case of Insulin/IGF-1 Signaling

**DOI:** 10.3390/cells13030288

**Published:** 2024-02-05

**Authors:** Paola Fabrizio, Allan Alcolei, Florence Solari

**Affiliations:** 1Laboratoire de Biologie et Modélisation de la Cellule, Ecole Normale Supérieure de Lyon, CNRS UMR5239, INSERM 1210, University Claude Bernard Lyon 1, 69364 Lyon, France; paola.fabrizio@ens-lyon.fr; 2INMG, MeLiS, CNRS UMR 5284, INSERM U1314, University Claude Bernard Lyon 1, 69008 Lyon, France; allan.alcolei@univ-lyon1.fr

**Keywords:** aging, tissue-specific, *C. elegans*, DAF-2, insulin-IGF-1 receptor, temporal resolution

## Abstract

The aging process is inherently complex, involving multiple mechanisms that interact at different biological scales. The nematode *Caenorhabditis elegans* is a simple model organism that has played a pivotal role in aging research following the discovery of mutations extending lifespan. Longevity pathways identified in *C. elegans* were subsequently found to be conserved and regulate lifespan in multiple species. These pathways intersect with fundamental hallmarks of aging that include nutrient sensing, epigenetic alterations, proteostasis loss, and mitochondrial dysfunction. Here we summarize recent data obtained in *C. elegans* highlighting the importance of studying aging at both the tissue and temporal scale. We then focus on the neuromuscular system to illustrate the kinetics of changes that take place with age. We describe recently developed tools that enabled the dissection of the contribution of the insulin/IGF-1 receptor ortholog DAF-2 to the regulation of worm mobility in specific tissues and at different ages. We also discuss guidelines and potential pitfalls in the use of these new tools. We further highlight the opportunities that they present, especially when combined with recent transcriptomic data, to address and resolve the inherent complexity of aging. Understanding how different aging processes interact within and between tissues at different life stages could ultimately suggest potential intervention points for age-related diseases.

## 1. Introduction

*C. elegans* has played an instrumental role in understanding the aging process since the late 80s when the first mutations that extend lifespan were discovered [1,2,3]. Subsequently, several longevity pathways identified in *C. elegans* were shown to play a conserved role in lifespan regulation of other organisms including, possibly, humans [4,5]. These pathways typically control molecular and cellular mechanisms recognized as the “hallmarks of aging”, which *C. elegans* has contributed significantly to defining (Figure 1). These include epigenetic alterations, loss of proteostasis, deactivated macroautophagy, deregulated nutrient sensing, mitochondrial dysfunction, and impaired intercellular communication (for a review, see [6]). Some of the genes that modulate lifespan are also involved in age-related diseases in humans [7]. Consistently, advanced age is the main risk factor for many common diseases such as cancers, cardiovascular disorders, and neurodegeneration. The diversity of the mechanisms involved in aging also underscores the importance of aging research to understanding various biological and medical challenges.

So far, enormous progress has been made in identifying longevity genes and pathways. However, recent findings by our laboratory and others reveal how far we are from elucidating the complexity of aging and how a multitude of genes and mechanisms at multiple scales contribute to declines in function, health, and lifespan.

Numerous reports have demonstrated that aging involves both cell-autonomous and non-cell- autonomous mechanisms (for a review, see [8]). The maintenance of whole-body homeostasis, whether under normal or stressful conditions, necessitates effective communication between tissues to ensure the preservation of our physiological functions. This raises several questions. How do differentiated tissues, which perform specific tasks, integrate whole-body homeostasis demand at the subcellular level with age? Do all tissues age at the same pace? Which age-associated changes are “synchronized” between tissues? Which ones are tissue-specific? What are the molecular pathways involved? Thanks to the development of an innovative tissue-specific methodology we started addressing these questions. Resolving this complexity is not a trivial matter, as preventing age-related diseases involves knowing when and where interventions should take place and how they may impact the body as a whole.

## 2. Increased Longevity Does Not Guarantee Improved Health in Old Age

Since the identification of *daf-2* and *age-1* (coding for the insulin/IGF-1 receptor and the catalytic subunit of the mammalian PI3Kinase, respectively) as conserved regulators of longevity, hundreds of genes have been found that modulate the lifespan of *C. elegans* (see GenAge database and [9]). Early efforts focused on high-throughput RNAi genetic screening for lifespan extension revealed important regulators such as mitochondrial genes [10,11,12]. Concomitantly, *C. elegans* emerged as a prominent model for assessing the impact of pharmaceutical interventions and small molecules on lifespan. Notably, some of these interventions demonstrated a conserved effect across other animal models, as documented in the DrugAge database.

The goal of these initial studies was to identify interventions that prolonged lifespan. However, longer lifespan is not always associated with an increase in healthy life expectancy [13,14,15,16]. “Healthy worms” are distinguished by their resilience to exogenous stressors, including oxidative, thermal, or endoplasmic reticulum-specific stress (Figure 1). In addition, their ability to retain mobility in late adulthood and maintain the pharyngeal pumping rate as they age often contributes to the definition of healthy life expectancy. It is important to note, however, that consensus metrics for defining “healthspan” in *C. elegans* [17] and within the broader field of geroscience research are still under discussion, as reported elsewhere [18].

The lack of overlap between lifespan and healthspan phenotypes underscores that the factors and/or tissues that limit worm survival do not entirely align with those required for maintaining optimal health in worms during old age. Thus, the lifespan phenotype is not sufficient to determine whether aging is accelerated or delayed.

Additionally, the phenotypes commonly studied in the context of aging, such as paralysis or stress survival, pose a challenge as they may stem from various cellular processes occurring in different tissues and at different ages. Therefore, it is imperative to discern the molecules and cellular pathways at play in each tissue during the aging process.

Several molecular mechanisms whose modulation impacts worm lifespan have been described, including the loss of proteostasis [19,20], mitochondrial dysfunction [21,22,23], dysregulated nutrient sensing [24,25], microbiome disruption [26,27], epigenetic alteration [28,29] and splicing deregulation [30]. The physiological decline of *C. elegans* is characterized by changes in mobility, pharyngeal pumping and offspring production [31], neuronal activity [32,33,34], and in stress resistance [35]. The sequence of molecular and cellular events at the tissue-scale and their impact on physiological decline remain essentially unknown (question mark).

### 2.1. Thinking about Aging at the Tissue-Scale

How do multiple aging mechanisms interact over time and space, i.e., within a tissue at different ages and between tissues of the same age? *C. elegans* is an optimal system for addressing these questions due to its short lifespan of only 20 days and its inherent transparency. This transparency enables the measurement of subcellular phenotypes with fine temporal resolution and in a tissue-specific manner, all within living animals. This is made possible through the use of fluorescent reporters, which provide a powerful tool for precise observation and analysis. Nevertheless, we still know very little about tissue-specific aging in *C. elegans*. Age-related modifications have been so far mostly observed on a whole-body scale. Those changes have been investigated at the molecular level by transcriptomic [36,37,38,39,40,41] and proteomic [42,43] analyses. Behavioral and functional alterations have also been reported such as decreased learning and memory [44], changes in the response properties of thermosensory neurons [45], or loss of mobility [31,46,47] (Figure 1).

Conversely, only a few tissue-specific changes have been investigated, such as those affecting the reproductive system, e.g., reproductive rate and number of offspring, germ cell deterioration, changes in oocyte morphology, and the appearance of tumorous germlines with age [48,49,50]. Other morphological changes have been reported for some neurons [51,52,53,54], as well as the loss of intestinal [55,56] or muscle cell integrity (e.g., sarcomere structure [46,57] and see below). A more detailed overview of the age-associated anatomical and physiological alterations in *C. elegans* is reported in two recent reviews [58,59] and illustrated in WormAtlas (https://www.wormatlas.org/aging/aginghomepage.htm (accessed on 1 December 2023)).

Despite these recent efforts, the study of tissue aging in *C. elegans* is still in its infancy, as we lack a true tissue-aging model that systematically describes the sequential changes that take place with age in each tissue (Figure 1).

### 2.2. Importance of Addressing Aging on a Temporal Scale

The importance of timing in longevity regulation was first suggested by the antagonistic pleiotropy theory of aging. This theory predicts that genes prolonging lifespan when inactivated during adulthood, have detrimental effects when inhibited earlier in development [60]. Such genes were identified in *C. elegans* using an RNAi screening strategy, focusing on 2700 genes essential for growth and development. Post-developmental inactivation of some of these genes revealed their role in lifespan regulation. An extended lifespan was observed for 64 of them belonging to different functional groups, such as translation, RNA, chromatin factors and metabolism [61]. This suggests that the constitutive inactivation of certain genes and mechanisms can mask their beneficial impact on lifespan, sometimes in unexpected ways, as described hereafter for autophagy.

The blockage of autophagy is considered one of the universal hallmarks of aging (Ref. [6] for review see [20,62]) (Figure 1). In agreement with this consensus, constitutive inactivation of autophagy reduces the lifespan of *C. elegans*, most likely due to the accumulation of defective proteins and organelles [63,64,65]. However, two studies in *C. elegans* have reported that inactivating autophagy can extend lifespan if the intervention takes place in adulthood or later [66,67]. Thus, autophagy might shift from a beneficial to a detrimental role in the context of age-associated dysfunction.

Conversely, the extension of lifespan can also be driven by changes occurring specifically during the early part of the life cycle. A paradigmatic example is the impact of the downregulation of certain mitochondrial genes.

Mitochondria are indispensable for key cellular functions, including ATP production, calcium homeostasis, and fatty acid metabolism. However, they also generate harmful reactive oxygen species (ROS) as a by-product of electron leakage from the electron transport chain (ETC). ROS are considered to play a causal role in aging according to the oxidative theory of aging [68].

Supporting this theory, there is evidence that the mild inactivation of components within the mitochondrial ETC prolongs lifespan in various organisms, including *C. elegans* [8,9,10,68,69], flies [70], and mice [71]. However, the temporal aspect of inactivation is of paramount importance in *C. elegans*. To exert a positive influence on lifespan, it is imperative to specifically inactivate mitochondrial genes during development rather than adulthood. Merkwirth et al. [72] later discovered that the observed extension of lifespan resulting from a modest inactivation of the ETC function involves an adaptative mitochondrial unfolded protein response (UPRmt). This response is orchestrated through the activation of epigenetic regulators, shedding light on the intricate mechanisms linking mitochondrial function, protein homeostasis, and epigenetics for the regulation of lifespan. In addition, UPRmt involves numerous effectors that have been extensively studied in *C. elegans* [73,74,75,76,77].

Overall, these observations underscore the critical importance of timing in the efficacy of life-prolonging interventions, a factor that can vary depending on the specific intervention or mechanism under consideration.

## 3. Decoupling Developmental Phenotypes and Lifespan: Insights from DAF-2/IIRc and LET-363/TOR

DAF-2/IIRc and LET-363/TOR stand out as two of the most extensively studied proteins with respect to their involvement in regulating longevity across numerous species. The serine-threonine kinase TOR is a central effector of lifespan extension by dietary restriction [78]. For many years, the effects of LET-363/TOR on *C. elegans* lifespan could not be studied rigorously for several reasons. The available mutants were arrested at the L3 larval stage, well before adulthood, and the inactivation of *let-363/TOR* by RNAi gave variable and confusing results. Subsequent investigations revealed that the initial ambiguity stemmed from the unintended knockdown of a mitochondrial gene, *mrpl-47*, through RNA interference. This gene is part of an operon with *let-363* and has a discernible impact on lifespan [12,79].

Strong *daf-2/IIRc* mutations also lead to early developmental phenotypes in *C. elegans*, including embryonic lethality and growth arrest. Weaker *daf-2/IIRc* inactivation is associated with both prolonged lifespan and fertility defects [80]. The fertility defects observed in *daf-2/IIRc* and *let-363/TOR* mutants [81] in *C. elegans* and other species have long been considered inseparable from their increased lifespan, in agreement with the disposable soma theory of aging [82,83]. Nevertheless, the development of new tools enabling the inducible degradation of protein by the AID (auxin inducible degradation) system [84] have shown that development and germline phenotypes can be uncoupled from the longevity phenotype for both proteins.

The AID system was discovered in plants and adapted to perform conditional protein depletion in *C. elegans*. It relies on the expression of a plant-specific F-box protein, TIR1, which is the substrate recognition component of an Skp1-Cullin-F-box (SCF) E3 ubiquitin ligase complex. TIR1 interacts with the substrate protein only in the presence of auxin and targets it for degradation by the proteasome. The introduction of a degron sequence by CRISPR/Cas9 genome editing into the gene encoding the protein to be degraded allows its recognition as a substrate by TIR1. However, this recognition occurs only in the presence of auxin, which can be introduced into the culture medium at any given point. Leveraging the tissue-specific expression of TIR1, the AID system provided a powerful tool to investigate the temporal and tissue-specific requisites of these proteins in the intricate regulation of lifespan.

The degradation of LET-363/TOR (or of its positive regulator RAGA-1) from the first day of adulthood extended the worms’ lifespan and thus makes it possible to dissociate increased lifespan from growth and reproductive disorders [85]. Inhibition of DAF-2/IIRc expression in adulthood alone also recapitulates the increased lifespan observed with the weak *daf-2/IIRc* allele without deleterious effects on development or fertility [16,86,87]. Remarkably, degrading DAF-2/IIRc at advanced age—when 50–75% of the population has already perished—results in a doubling of life expectancy among the remaining survivors. This intriguing finding suggests that even in the later stages of life, old worms retain the capacity to engage anti-aging mechanisms [87].

## 4. Age-Associated Subcellular Changes: Commonalities and Distinct Patterns across Diverse Tissues

The development of tissue-specific reporters in *C. elegans* showed that fundamental cellular processes such as autophagy [88,89] or protein degradation by the ubiquitin proteasome system (UPS) [90,91] exhibited variation with age. At the age of one week, *C. elegans* already displays severe symptoms of aging and stops reproducing. Analysis of autophagy flux provided evidence for a blockade occurring from this age in both neurons and muscle. UPS activity also showed a tissue-specific variation with age. However, while it was clearly reduced in some neurons, it remained unchanged in muscle cells as compared to young animals [90,91].

Maintaining protein homeostasis relies on a tissue-specific repertoire of chaperone proteins, and their expression exhibits age-related variations in both *C. elegans* and humans. Consequently, the response of a tissue to a homeostasis imbalance is shaped by its functional characteristics and the age of the individual [92,93,94].

Approaches allowing the expression or inactivation of specific genes in certain tissues have revealed the involvement of secondary signals and inter-tissue communication for the modulation of lifespan (for a review, see [8]). Coordinated responses at the whole-body scale have been demonstrated for the regulation of protein [95,96] and mitochondrial homeostasis [97,98,99,100]. Distant tissues can adopt phenotypes similar to the signaling tissue or alternatively, a different adaptive response may be triggered in distant tissues (for reviews, see [21,101]). This has largely been illustrated through the study of the unfolded protein response (of both ER and mitochondria) that can be triggered in the intestine or in muscle from neurons [98,102] or from the germline [103,104] through Wnt, neurotransmitter, and neuropeptides signaling.

DAF-2/IIRc and LET-363/TOR have also been shown to have tissue-specific functions in controlling lifespan during adulthood [16,85,87]. DAF-2/IIRc is ubiquitously expressed in adult worms [16]. However, the regulation of lifespan depends essentially on its activity in neurons and the intestine and involves interaction between these two tissues. Other tissues may be required for longevity regulation since ubiquitous *daf-2* inactivation is associated with a longer lifespan compared to combined *daf-2* inactivation in neuronal and gut tissues. Notably, while LET-363/TOR inactivation in neurons during adulthood is sufficient to extend lifespan, its ubiquitous depletion reduced it [85]. These results demonstrate that LET-363/TOR inhibition is deleterious in some tissues but advantageous in others, highlighting the importance of identifying the key tissue(s) in which a pathway acts to regulate aging rather than targeting it broadly throughout the entire body.

More recently, the impact of DAF-2/IIRc has been addressed in greater detail beyond the lifespan phenotype, examining its effects at the temporal and tissue scale, particularly in the regulation of muscle aging, as described hereafter.

## 5. Defining Sequential Changes Characterizing Muscle Aging in *C. elegans*: A Step Forward

*C. elegans* possesses striated muscle cells with a highly conserved structure of sarcomeric contractile units that support the worm’s locomotion. A stereotyped sequence of molecular and cellular events characterizes muscle aging. These changes include: (1) a sharp decrease in sarcomeric transcripts in the first days of adulthood [105], followed by (2) the progressive fragmentation of muscle mitochondria [105,106], and (3) a blockade of autophagy [88,105]. Notably, the mRNA levels of 27 sarcomeric genes show a marked downregulation, ranging from 5 to 20 times, within the first week of adulthood, starting as early as day 1 of adulthood.

Based on promoter sequence analysis, Mergoud et al. [105] identified three conserved myogenic transcription factors as potential regulators: the conserved MAD-box UNC-120/SRF, the basic helix-loop-helix HLH-1/MYOD, and HND-1/HAND1 transcription factors, all known for their role in embryonic muscle differentiation [107]. They showed that only *unc-120* and *hlh-1* are expressed during adulthood. Inactivating *unc-120* expression during adulthood only accelerated the loss of mobility, the decline of muscle transcripts, the fragmentation of mitochondria, and the blockade of autophagy. Conversely, inactivating *hlh-1* did not affect the kinetics of any of the above biomarkers. Furthermore, the overexpression of UNC-120/SRF in muscle only was sufficient to delay the appearance of muscle aging biomarkers and improved mobility in aged animals. Importantly, while *unc-120* inactivation reduced lifespan, the beneficial impact of UNC-120 muscle overexpression on muscle aging was not associated with lifespan extension.

### 5.1. Delayed Age-Associated Sub-Cellular Changes through DAF-2/IIRc Inactivation

DAF-2/IIRc inactivation increases worm lifespan by 100% and also prevents a plethora of age-related changes, including vulnerability to stress and the loss of mobility associated with aging. Li et al. [33] observed a decline in neurosecretion from motor neurons due to aging, commencing at day 7 of adulthood. This decline appeared mitigated in aged *daf-2* mutants, indicating that the loss of DAF-2/IIRc may enhance locomotion, at least partially, by preserving motoneuron function. *daf-2* mutants compared to wild-type worms also show a delay in the downregulation of sarcomeric transcripts, muscle mitochondria fragmentation, and the blockade of autophagy [88,105].

Hence, the UNC-120/SRF transcription factor and the DAF-2/IIRc receptor both impact the entire sequence of muscle changes. While *unc-120* inactivation accelerates this sequence, muscle *UNC-120* OE or *daf-2* inactivation has the opposite effect, delaying it. This suggests a potential interconnection between the markers of muscle aging that appear at different ages, which is consistent with the concept of the temporal scaling of *C. elegans* aging proposed by Stroustrup et al. [108]. These authors observed that interventions as diverse as changes in diet, temperature, exposure to oxidative stress, and disruption of different genes (the heat shock factor *hsf-1*, the hypoxia-inducible factor *hif-1*, and the insulin/IGF-1 pathway components *daf-2*, *age-1*, and *daf-16*) all alter lifespan distributions by an apparent stretching or shrinking of time. They thus conclude that organismal aging dynamics are invariant across genetic and environmental contexts [108]. However, later works showed that the underlying regulatory mechanisms are more complex than expected (see below).

### 5.2. DAF-2/IIRc: Orchestrating Age-Dependent Mobility through Dual Autonomous and Non-Autonomous Pathways

Roy et al. [16] constructed a *daf-2* knock-in line to visualize and deplete the DAF-2/IIRc protein in a spatially and temporally controlled manner using the AID system. As expected, reducing DAF-2/IIRc expression in the nervous system or in the intestine during adulthood only, is sufficient to extend lifespan [16,87,109,110,111], although it does not preserve mobility at middle age [16] (Figure 2).

Mobility relies on the functional coordination of muscle and neuronal tissues. The selective downregulation of DAF-2/IIRc in muscles had no impact on lifespan or resistance to oxidative stress. Interestingly this targeted downregulation resulted in an increase in mobility in middle-aged (13-day-old) adults. The enhanced mobility was comparable to the effect observed with the weak *daf-2* allele.

The canonical DAF-2/IIRc pathway involves the conserved FOXO transcription factor DAF-16, the activation of which is essential for the extension of lifespan. However, the regulation of mobility by muscle DAF-2/IIRc does not depend on DAF-16/FOXO, but requires UNC-120/SRF.

Furthermore, the neuronal inactivation of DAF-2/IIRc unexpectedly reduces mobility, but only in early adulthood and via the DAF-16/FOXO transcription factor (Figure 2).

These results show that the longer lifespan and preserved mobility phenotype observed in *daf-2* mutants are in fact the sum of independent effects in different tissues and via tissue-specific effectors.

In *C. elegans*, the age-associated regulation of mobility by DAF-2/IIRc is determined by the sequential and opposing impact of neurons and muscle and can be dissociated from the lifespan phenotype. Intestinal and neuronal DAF-2/IIRc activities modulate lifespan, whereas muscle DAF-2/IIRc does not. Neuronal DAF-2/IIRc promotes mobility in early adulthood through the inhibition of DAF-16/FOXO, whereas muscle DAF-2/IIRc decreases mobility in middle age through the inactivation of UNC-120/SRF.

Overall, Roy et al. demonstrated that the wild-type DAF-2/DAF-16 signaling pathway plays a crucial role in promoting the mobility of young worms within neurons. However, as worms age, this positive influence diminishes. Conversely, the muscular expression of DAF-2/IIRc seems to have no impact on early age mobility, while it impedes older worms’ mobility through the inactivation of UNC-120/SRF. Roy et al. [16] also showed that both neuronal and muscular DAF-2/IIRc activities promote the fragmentation of muscle mitochondria with age.

*daf-2* mutations were shown to prevent a number of behavioral, molecular, and cellular phenotypes associated with aging [112]. In order to address the physiological relationship between those different phenotypes, it will be important to determine whether these phenotypes depend on specific *daf-2* activities in tissues, and at what age *daf-2* activity is required for their control.

## 6. Deciphering Sequential Changes in Tissue Aging in *C. elegans*: Limits and Future Directions

Taken together, the analysis of *daf-2* mutants reveals that integrated phenotypes such as longevity or mobility can result from the sum of opposing age-dependent and tissue-specific activities. Obtaining a complete picture of the molecular and cellular bases of aging and their interactions, even in a simple organism like *C. elegans*, may seem an unattainable goal when considering its multifactorial nature. To this end, we propose to use a more systematic analysis of aging at the tissue level.

Recent tools allowing tissue-specific expression or inactivation in *C. elegans* will contribute greatly to the investigation of the kinetics of age-related changes and their regulation in different tissues. Nevertheless, care must be taken when using transgenic tools, as discussed here. Early transgenesis techniques in worms involved the expression of a transgene (non-integrated or integrated into the genome after irradiation) in multiple copies. Thus, in numerous studies published so far, the authors used animals expressing high levels of transgenic proteins which may interfere with physiological aging. As an example, the most widely used biomarker of muscle aging is the MYO-3::GFP translational reporter, which overexpresses the MYO-3 protein (myosin heavy chain) fused to GFP. This line has a shortened lifespan compared to wild-type animals (F. Solari, pers. com. [113]) and shows fiber disorganization well before the adult stage [105,114]. The same translational fusion, introduced by knocking GFP at the endogenous *myo-3* locus (by CRISPR/Cas9 strategy), revealed that muscle fiber did not display the disorganization described above, even at an age when 50% of the worms were no longer moving [105]. Another example is the strain used to visualize mitochondrial morphology, which relies on mitochondria-targeted GFP. When this transgene is expressed in a multicopy array, mitochondrial fragmentation begins as early as the first week of worm adulthood [105,106]. On the other hand, when the same reporter is expressed from a single-copy transgene, fragmentation begins at a later age [16].

In summary, it is advisable to prioritize single-copy transgenic lines and verify that their lifespan, along with other pertinent phenotypes, remains relatively stable compared to wild-type animals. This approach proves to be a sound starting point for the study of aging at the tissue level.

The use of the AID system in *C. elegans,* which enables the time- and tissue-dependent inactivation of a given protein, also has certain limitations. Auxin at 1 mM concentration has been reported to promote ER stress resistance and other phenotypes in wild-type animals [115]. However, a lower concentration may induce efficient degradation without triggering additional effects as shown for LET-363/TOR [85]. So, one should carefully control the impact of auxin on their favorite phenotype when using this experimental strategy.

One of the main advantages of *C. elegans* is that all its tissues are post-mitotic in the adult (with the exception of the germ line) so that the age of a cell corresponds to the age of the whole body. Dynamic gene expression can thus be monitored without the confounding effect of a change in tissue cell composition, as has been reported in mammals with age [116,117]. Various C. *elegans* repositories containing genes whose expression is enriched in specific tissues will be instrumental in defining the key molecular players required for the functional status of a given tissue. These players may include shared elements among tissues, such as the products of house-keeping genes, as well as those specific to each individual tissue. Repository data include transcriptomic results obtained through bulk analysis [118] and at the single cell resolution of young adult animals ([119] and see WormSeq website). The same results have also been analyzed using the cell2cell tool to deduce potential cell-to-cell interactions based on the expression of ligand- and receptor-encoding genes across cells in single-cell transcriptomic datasets [118]. More recently, three independent laboratories reported variations in gene expression in *C. elegans* concerning age and different genetic backgrounds, observed at the resolution of single cells or nuclei [120,121,122]. Those studies identified a few common biological pathways that are affected by age in several tissues. These encompass genes linked to translation and DNA repair, occasionally exhibiting opposite patterns. Most age-related changes seem to be tissue-specific. It is crucial to validate those variations in living animals and to ascertain whether these transcriptional changes influence corresponding protein levels and functions as the organism ages.

## 7. Conclusions

The implementation of tissue-specific strategies, coupled with temporal resolution, will enable us to answer fundamental questions about aging. Are certain tissues more susceptible to early signs of cellular aging? Do these tissues act as drivers of organismal aging through secondary signaling mechanisms? These strategies are expected to make a significant contribution to our understanding of the aging process at the cellular and tissue levels, unraveling the intricate integration that leads to the complexities of organismal aging.

## Figures and Tables

**Figure 1 cells-13-00288-f001:**
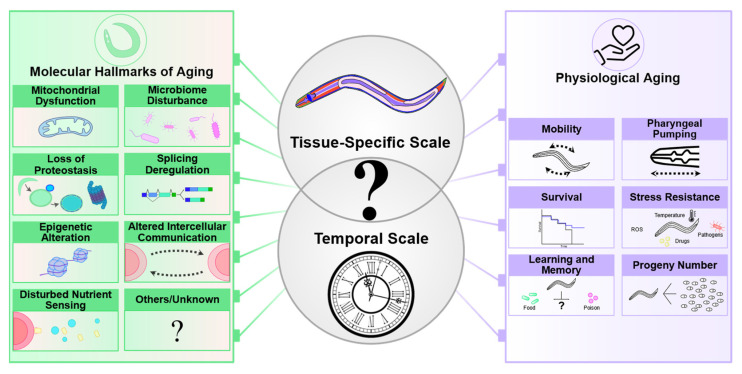
Addressing aging at different scales in *C. elegans*.

**Figure 2 cells-13-00288-f002:**
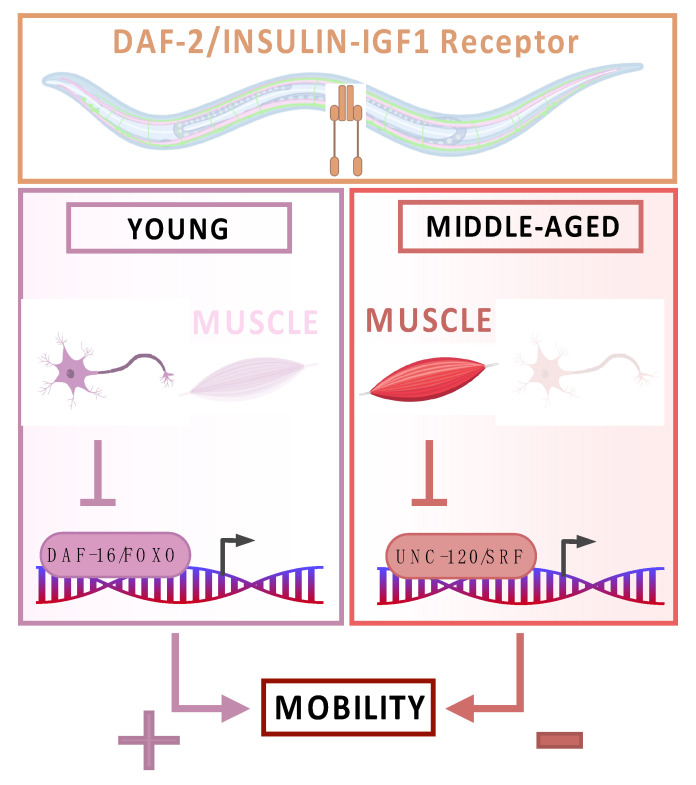
Tissue-specific regulation of lifespan and mobility with age by the DAF-2 insulin-IGF-1 receptor.

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
