# Peer review of "Considering Caenorhabditis elegans Aging on a Temporal and Tissue Scale: The Case of Insulin/IGF-1 Signaling"

_cells, 2024, doi:10.3390/cells13030288_

Round 1

Reviewer 1 Report

Comments and Suggestions for Authors

This is a nice (although very comprehensive) review addressing critical aspects which need to be considered when assessing aging at the organismal level. Namely, temporal and tissue resolutions (dosage should be also considered) are not always carefully taken into consideration when addressing the impact of interventions on C. elegans health and lifespan, which instead may lead to different and important discoveries.

Minor points should be addressed before publication.

1) The authors should carefully revise the manuscript in search of typos

2) The review could be reorganized with subchapters to better reflect the different topics they discussed. For instance:

- chapter 3 and 4 could actually be 2.1 and 2.2

- chapter 5 to 8 are all primarily about daf-2 pathways and it would be easy to follow the manuscript if they are under the same heading (eg chapter 3) with different sub-chapter (3.1, 3.2,…..)

3) Rather than a review this is more a perspective/commentary manuscript. To be a review they should include much more extensive information on different age-modulating signaling. Alternatively, they should rewrite the title to focus on the IGF/insulin signaling. For instance: “Considering Caenorhabditis elegans aging on a temporal and tissue scale: the case of IGF/insulin mutants or signalling”

4) In chapter 4 the authors bring two important examples of how age-regulatory pathways should be considered from different perspectives (temporal, tissue and dosage), namely, autophagy and mitochondrial signalling. However, for both pathways, additional studies should be cited to more comprehensively cover the topic, especially related to dosage and tissue dependency.

Author Response

Dear Cells Editors,

Herein we submit a revised version of our review article entitled “Considering Caenorhabditis elegans aging on a temporal and tissue scale: the case of insulin/IGF-1 signaling” by Paola Fabrizio, Allan Alcolei and myself. We have addressed all the concerns raised by the two reviewers and we are providing point-to-point responses to their reviews. 

We hope this revised version is now suitable for publication in Cells.

Sincerely,

Florence Solari, PhD

Reviewer 1

This is a nice (although very comprehensive) review addressing critical aspects which need to be considered when assessing aging at the organismal level. Namely, temporal and tissue resolutions (dosage should be also considered) are not always carefully taken into consideration when addressing the impact of interventions on C. elegans health and lifespan, which instead may lead to different and important discoveries.

Minor points should be addressed before publication. 

  • The authors should carefully revise the manuscript in search of typos

=>We carefully looked for typos in the manuscript and removed them.

2) The review could be reorganized with subchapters to better reflect the different topics they discussed. For instance:

- chapter 3 and 4 could actually be 2.1 and 2.2

- chapter 5 to 8 are all primarily about daf-2 pathways and it would be easy to follow the manuscript if they are under the same heading (eg chapter 3) with different sub-chapter (3.1, 3.2,…..)

 => We followed the advice of the reviewer. However, we preferred to keep separate (chapter 5) the paragraphs where we discussed more specifically the age-dependent changes occurring in muscle and how daf-2 and unc-120 affect them.

3) Rather than a review this is more a perspective/commentary manuscript. To be a review they should include much more extensive information on different age-modulating signaling. Alternatively, they should rewrite the title to focus on the IGF/insulin signaling. For instance: “Considering Caenorhabditis elegans aging on a temporal and tissue scale: the case of IGF/insulin mutants or signalling”

  => We changed the title according to the reviewer’s suggestion.

4) In chapter 4 the authors bring two important examples of how age-regulatory pathways should be considered from different perspectives (temporal, tissue and dosage), namely, autophagy and mitochondrial signalling. However, for both pathways, additional studies should be cited to more comprehensively cover the topic, especially related to dosage and tissue dependency.

=> Several references have been added (in red) accordingly: lines 135, 150, 158, 221

Reviewer 2 Report

Comments and Suggestions for Authors

In this manuscript entitled “Considering Caenorhabditis elegans aging on a temporal and tissue scale”, the authors reviewed the recent research studying aging at both tissue- and temporal- scale strategies in C. elegans. This paper highlights the recent tools allowing tissue-specific expression or inactivation, including tissue-specific gene expression and the use of the AID system. They investigated the nematode C. elegans as a model organism, that has played a pivotal role in aging research following the discovery of mutations extending lifespan. Notably, the study suggests the significant tissue-specific strategies with temporal resolution to understand organismal aging at the cellular and tissue levels.

Major comments

1.    I recommend that the authors add multiple citations, including the following ones. Add paper that describe the roles of DAF-16 in organismal aging (Lee and Lee, 2022, PMID: 36380728). For describing the roles of autophagy in aging, please add citations describing the link between calorie restriction and autophagy (Park et al., 2022, PMID: 34949740)

2.    It will be better to provide high resolution figures throughout the manuscript.

Minor comments

1. Please correct typos throughout the manuscript. Following are some examples.

1)    On page 1, line 30, please correct “hallmarks of aging>> to “hallmarks of aging”

2)    On page 3, line 108, please correct “transcription  [36-41]” to “transcription [36-41]”

3)    On page 6, line 236, please correct “DAF-2//IIRc” to “DAF-2/IIRc”

4)    On page 8, line 304, please correct “Figure 2 =” to “Figure 2”

5)    On page 9, line 372, please correct “cell2cell” to “cell-to-cell”

Comments on the Quality of English Language

None

Author Response

Dear Cells Editors,

Herein we submit a revised version of our review article entitled “Considering Caenorhabditis elegans aging on a temporal and tissue scale: the case of insulin/IGF-1 signaling” by Paola Fabrizio, Allan Alcolei and myself. We have addressed all the concerns raised by the two reviewers and we are providing point-to-point responses to their reviews. 

We hope this revised version is now suitable for publication in Cells.

Sincerely,

Florence Solari, PhD

Reviewer 2

In this manuscript entitled “Considering Caenorhabditis elegans aging on a temporal and tissue scale”, the authors reviewed the recent research studying aging at both tissue- and temporal- scale strategies in C. elegans. This paper highlights the recent tools allowing tissue-specific expression or inactivation, including tissue-specific gene expression and the use of the AID system. They investigated the nematode C. elegans as a model organism, that has played a pivotal role in aging research following the discovery of mutations extending lifespan. Notably, the study suggests the significant tissue-specific strategies with temporal resolution to understand organismal aging at the cellular and tissue levels.

Major comments

  1. I recommend that the authors add multiple citations, including the following ones.

 Add paper that describe the roles of DAF-16 in organismal aging (Lee and Lee, 2022, PMID: 36380728). For describing the roles of autophagy in aging, please add citations describing the link between calorie restriction and autophagy (Park et al., 2022, PMID: 34949740)

=>Reference 1 is now cited in line 325, and reference 2 is cited in line 131.

  1. It will be better to provide high resolution figures throughout the manuscript.

=> We are now presenting high resolution figures.

Minor comments

  1. Please correct typos throughout the manuscript. Following are some examples.

1)    On page 1, line 30, please correct “hallmarks of aging>> to “hallmarks of aging”

2)    On page 3, line 108, please correct “transcription  [36-41]” to “transcription [36-41]”

3)    On page 6, line 236, please correct “DAF-2//IIRc” to “DAF-2/IIRc”

4)    On page 8, line 304, please correct “Figure 2 =” to “Figure 2”

5)    On page 9, line 372, please correct “cell2cell” to “cell-to-cell”

=> All corrected except “cell2cell” which is the correct name of the tool used (Armingol et al., Plos Compu Biol, 2022)